# How effective were Australian Quarantine Stations in mitigating transmission aboard ships during the influenza pandemic of 1918-19?

**Punya Alahakoon**[1,3,4], **Peter G. Taylor**[1], **James M. McCaw**[1,2]*

**1** School of Mathematics and Statistics,The University of Melbourne, Melbourne, Australia, **2** Centre for Epidemiology and Biostatistics, Melbourne School of Population and Global Health, The University of Melbourne, Melbourne, Australia, **3** School of Population Health, University of New South Wales, Sydney, Australia, **4** Kirby Institute, University of New South Wales, Sydney, Australia

* jamesm@unimelb.edu.au

**Data Availability Statement:** Details on the available data and related codes are publicly available and included in the supplementary

## Abstract

The influenza pandemic of 1918-19 was the most devastating pandemic of the 20th century. It killed an estimated 50–100 million people worldwide. In late 1918, when the severity of the disease was apparent, the Australian Quarantine Service was established. Vessels returning from overseas and inter-state were intercepted, and people were examined for signs of illness and quarantined. Some of these vessels carried the infection throughout their voyage and cases were prevalent by the time the ship arrived at a Quarantine Station. We study four outbreaks that took place on board the *Medic, Boonah, Devon,* and *Manuka* in late 1918. These ships had returned from overseas and some of them were carrying troops that served in the First World War. By analysing these outbreaks under a stochastic Bayesian hierarchical modeling framework, we estimate the transmission rates among crew and passengers aboard these ships. Furthermore, we ask whether the removal of infectious, convalescent, and healthy individuals after arriving at a Quarantine Station in Australia was an effective public health response.

## Author summary

The influenza pandemic of 1918–19 was one of the deadliest pandemics in history. In Australia, when it was apparent that the virus was severe, a quarantine service was established to intercept and quarantine ships that returned from overseas and travelled inter-state. In this study, we look at the records of outbreaks on board the *Medic, Boonah, Devon*, and *Manuka*. Some of the ships carried surviving troops from the First World War, and infections were prevalent when they arrived at a quarantine station. Infectious, convalescent, and healthy individuals on board were moved to the quarantine station for treatment or isolation. We model the outbreaks on the four ships using stochastic epidemic models and estimate the model parameters within a hierarchical framework.

materials and on GitHub via https://github.com/PunyaAlahakoon/Ship_outbreaks_1918.git.

**Funding:** P.A. was supported by a Melbourne Research Scholarship from the University of Melbourne. P.G.T. would like to acknowledge the support of the Australian Research Council via the Centre of Excellence for Mathematical and Statistical Frontiers (ACEMS). The funders had no role in study design, data collection and analysis, decision to publish, or preparation of the manuscript.

**Competing interests:** The authors have declared that no competing interests exist.

Furthermore, we investigate whether the removal of individuals with various disease states was an effective intervention measure from a public health perspective.

## 1 Introduction

The influenza pandemic of 1918–19 was the most devastating pandemic in the 20th century [1–5]. The pandemic spread across the world in three consecutive waves: March 1918, September–November 1918, and early 1919 [6]. It has been estimated that the pandemic infected one-third of the world's population and killed an estimated 50–100 million people worldwide [6, 7] with a relatively high mortality rate among young adults [8, 9]. These estimated figures may not be accurate as the recorded number of infections and deaths may be an understatement due to "misdiagnosis, non-registration, missing records, and non-medical certification" [10].

Caused by an H1N1 influenza virus with an avian gene history, it is commonly believed that the 1918 influenza strain may have originated in the United States, spread through army camps, and eventually across the world [6] via troop ships and battlefields [3], coinciding with the end of First World War [11]. The spread of the disease beyond port cities may have been further amplified by local transport networks such as railways [3, 7]. The high levels of global population mixing may have contributed to the rapid global spread of the disease, along with other societal disadvantages such as poverty and overcrowding [12]. Studies of influenza data from cities in England and Wales and the United States army camps suggest that prior immunity from exposure to other influenza-like diseases among the older age groups in urban areas (and not isolated and disadvantaged countries and communities) may have reduced pandemic attack rates [13]. Furthermore, the waning of that immunity [13], changes in the phenotype virus [14], immunity acquired through previous waves [15], behavioural changes, temperature changes and school closures [16] all may have contributed to the multi-wave behaviour of the pandemic. During the first wave of the pandemic, the disease was of a mild character and the mortality rates were not unusual although a large number of young adults were still affected [7, 10]. The second wave was more virulent and infected patients suffered from high fever, cyanosis, and fluid accumulation in the lungs. When it was apparent that the 1918 influenza strain was severe, many countries adopted interventions to limit the spread. Some of these included use of face masks, improved hygiene, limiting of mass gatherings, and imposing strict maritime quarantine measures [7].

Maritime quarantine measures were implemented by many countries in 1918. However, these measures were generally unsuccessful as the restrictions were imposed too late, quarantine measures were breached, or the virus was already circulating in the community [7]. Australia, however, remained an exception. Their quarantine measures were successful in protecting Australians from the second wave until December 1918 when the measures were finally breached [5, 17]. Furthermore, it has been argued that Australia's quarantine measures indirectly helped in protecting Pacific Islands that depended on Australian supply ships [7, 18–21].

Apart from its successful quarantine measures, Australia's maritime isolation from the rest of the world also contributed to the delay of the pandemic reaching Australia. This was helpful in understanding the nature of the disease, learning which countries were mostly impacted by the pandemic, and also gave time for preparation of the quarantine officers to impose additional quarantine measures on vessels that had influenza during their voyages. This strategy was led by J. H. L. Cumpston, Director of Quarantine, who later became Australias first Commonwealth Director-General of Health [22, 23]. This quarantine programme started in early

October 1918 and lasted for six months until April 1919. Through this service, 228 incoming vessels, overseas or interstate, were intercepted [22, 23]. Medical officers on board were required to record occurrences of infections that took place during the ship's voyages. Quarantine Stations, located near most ports in Australia, also documented infections within the Quarantine Station and aboard a ship once it had initiated its quarantine period. This resulted in a service publication [23] that contains extensive details of the dynamics of outbreaks that took place on board ships which form the focus of this study.

We studied the outbreaks of influenza that occurred in late 1918 that took place on board the *Medic, Boonah, Devon*, and *Manuka*. The *Medic, Boonah*, and *Devon* transported Australian troops who served in influenza-affected parts of the world during the First World War [22, 23]. While *Medic* and *Boonah* recorded the removal of infectious, convalescent, and healthy individuals after arriving at the first Australian Quarantine Station, *Devon* and *Manuka* did not record these removals. We were primarily interested in estimating the key epidemiological parameters associated with these outbreaks that can provide improved insights to influenza outbreaks on board ships. Furthermore, we investigated whether Australian Quarantine Stations were beneficial in mitigating transmission on board these ships.

Consequently, we modeled each outbreak using a ship-specific stochastic two-group (crew and passenger) metapopulation model that captured the dynamics of a ship's journey and its time at an Australian Quarantine Station until the outbreak died out. We used a Bayesian hierarchical modeling approach to estimate model parameters. We demonstrated that generally, in the ships that transported troops, the transmission rates within crew and passengers (civilians and troops) were higher than those between the crew and passengers. We further calculated that the basic reproduction numbers, $R_0$, for *Medic, Boonah, Devon,* and *Manuka* were 6.38 (median) [(3.96, 11.06) quartile], 2.53 [2.07, 3.25], 4.46 [2.78, 8.97], and 2.45 [1.87, 3.96] respectively. These numbers were generally high and consistent with most settings where the populations were closed and isolated. Furthermore, we demonstrated through counterfactual analysis, that, the removal of individuals from aboard the *Medic* reduced the spread of infection. If all persons had remained aboard, we estimated an additional 18 (median) [(5, 31) quartile range] infections would have occurred. In contrast, removals from the *Boonah* were estimated to have had a negligible impact on the outbreak aboard the vessel. Overall, we concluded that Quarantine Stations in Australia provided varied benefits, depending on the epidemiological status of the ship at the time of quarantine.

## 2 Materials and methods

### 2.1 A ship's journey

The common characteristics of the Australian ships and their journey is as follows. As influenza and other infectious diseases were prevalent during this time, some Australian ships already had hospitals on board designed specifically to treat infectious individuals. In some situations when hospital accommodation was inadequate or absent, isolation areas were introduced during the ship's journey. On some occasions, infectious individuals were disembarked at overseas ports and the rest of the ship's population continued on their way to Australia. Furthermore, some of the ships' recorded cases of disease were not confirmed as influenza although they had influenza-like symptoms (high fever, pneumonia, etc.). These unconfirmed cases may be considered as individuals who showed fever-like symptoms. Some of the passengers, mostly troops, were also vaccinated before or after their journey. However, the vaccine, developed by the Commonwealth Serum Laboratories (CSL) in Australia, was a mixed bacterial vaccine (the influenza virus was unknown to science at the time) and there is no evidence

that it was effective in reducing transmission [24]. Some evaluations suggested that the vaccines were partially effective in preventing deaths [23].

When the ship reached an Australian port, the ship's population was required to perform a seven or three day quarantine period depending on the nature of the infection, prevalent cases at the time of arrival in Australia, and the port of presumed source of infection. On arrival at the Australian port, infectious individuals, if present, were taken to the Quarantine Station hospital to be treated. Daily thermometer parades were held on the remaining ship's population and any person showing an elevated temperature was separately isolated for observation. On some occasions, other intervention measures such as inoculation with the vaccine or inhalation of a zinc sulphate solution were undertaken. There is no evidence that zinc sulphate (or opium, salt water, alcohol) was effective in reducing transmission [7]. If the ship did not reach its final destination at the first Quarantine Station, the remaining personnel and the ship sailed in quarantine to the next Australian port where the same quarantine measures were taken on the remaining population of the ship. This quarantine procedure was taken until the ship reached its final destination.

### 2.2 Data

In the service publication by [23], individuals aboard seventy-nine ships were identified to have acquired influenza prior to or during an overseas voyage. Ninety-two outbreaks were recorded as influenza acquisition via inter-state travel. There were also records of ships that did not record any infections during their voyages.

For each of the seventy-nine ships that travelled overseas and on which passengers and/or crew acquired influenza, we first examined each ships record to identify those that reported a major outbreak during their voyage. As a threshold, we chose only those ships that reported at least 20 cases during the voyage (22 ships). Of those 22 ships, most records had key information such as the total epidemic size (grouped by crew, civilian, and/or troop), total population size, source of infection, and port of arrival in Australia. However, only 11 of the 22 ship records included daily prevalence or incidence data, which is necessary for conducting a time-series analysis. The records for five of these 11 ships, while describing the time-series in the textual description of the outbreak(s), did not record the total number of infections by group. A further two ships had outbreaks that were recorded as only having taken place among one of the three groups (crew, civilian, troops), so were excluded from analyses given concern that data on infections in other groups may be systematically missing. This left four of the 79 ships available for analysis (*Boonah*, *Medic*, *Devon* and *Manuka*). See Fig 1 for a summary of this outbreak selection procedure.

These four shipboard outbreaks had detailed time-series data on daily new case numbers, daily prevalence, incidence, the final size of the epidemic and sufficient qualitative information to formulate a suitable transmission model for the outbreak.

Of the *Medic*'s 989 population, there were 829 troops. The rest of the population consisted of 4 civilians and 156 crew. *Boonah*'s population consisted of 916 troops and 164 crew, a total of 1080. Of the *Devon*'s 1096 population, 920 were troops, 66 were civilians, and 110 were passengers. *Manuka*'s record does not mention the transportation of troops. Of its population of 203, 95 were crew and 108 were assumed to be civilians. The records of *Medic*, *Boonah*, and *Manuka* document the prevalence of cases up to the time they arrived at the quarantine stations. Out of them, only *Medic* and *Boonah* describe the removal of infectious, infected, and healthy people from the ships to be quarantined and/or treated. The record of *Medic* also mentions the removal of mild cases (see S1 Text for details of these data). Table 1 provides a summary of the details of each ship's population, voyage, quarantine details, and outbreak

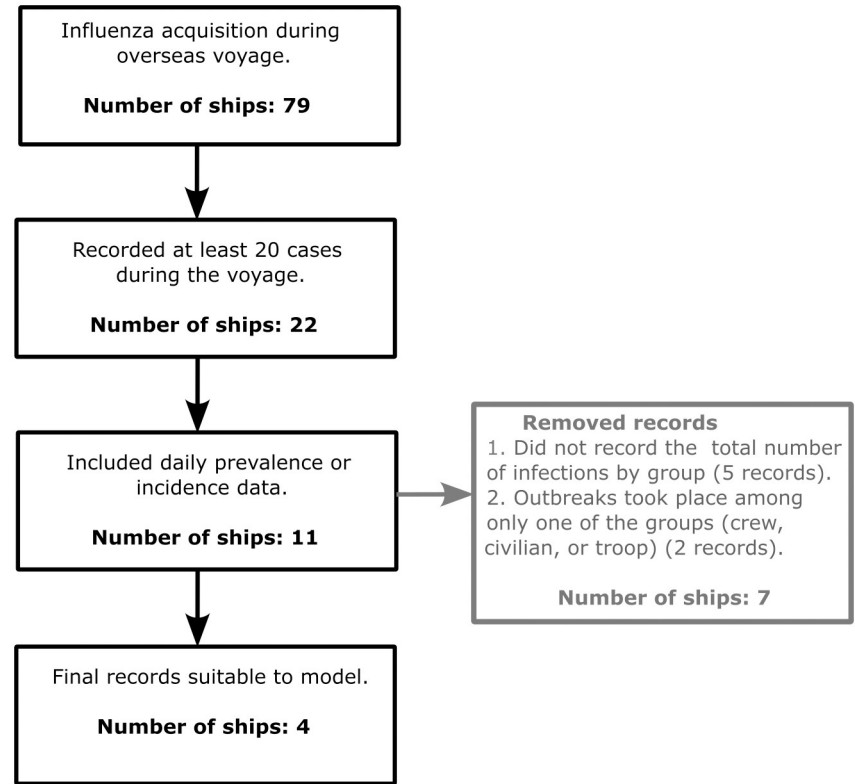

**Fig 1. Attrition diagram that presents the process of choosing four outbreaks to study.**

characteristics. Fig 2 shows a snapshot of the data available for the *Medic*. S1 Text presents illustrations for the other three ships and includes details of the GitHub repository where annotations and assumptions for each ship record can be found.

For all four ships, the data mostly consisted of the number of new infections that occurred during the time period of the voyages and their quarantine periods. For some outbreaks, the time-series data did not contain the details of the newly infected person's group (that is, whether they were a crew member or a passenger). However, for all four outbreaks, the total number of infections among each group and the total number of deaths due to influenza were recorded. S1 Text provides complete details of the recorded data we have gathered.

**Table 1. A summary of the ships' details.**

| Ship | Crew size | Passenger size (Civilian and/or troops) | Departure date at the port of presumed source of infection | Port of presumed source of infection | Arrival date in Australia | First arrived Quarantine Station | Total number of infections | Total number of deaths |
|---|---|---|---|---|---|---|---|---|
| *Medic* | 156 | 833 | Nov 11, 1918 | Wellington, New Zealand | Nov 21, 1918 | Sydney | 313 | 23 |
| *Boonah* | 164 | 916 | Nov 24, 1918 | Durban, South Africa | Dec 11, 1918 | Fremantle | 470 | 16 |
| *Devon* | 110 | 986 | Oct 13, 1918 | Suez, Egypt | Nov 15, 1918 | Fremantle | 95 | 0 |
| *Manuka* | 95 | 108 | Nov 7, 1918 | Wellington, New Zealand | Nov 13, 1918 | Sydney | 41 | 1 |

**(A)**

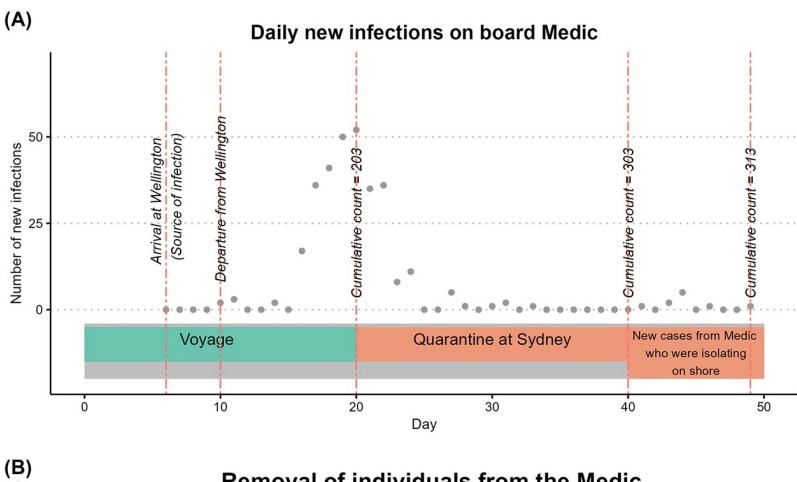

**(B)**

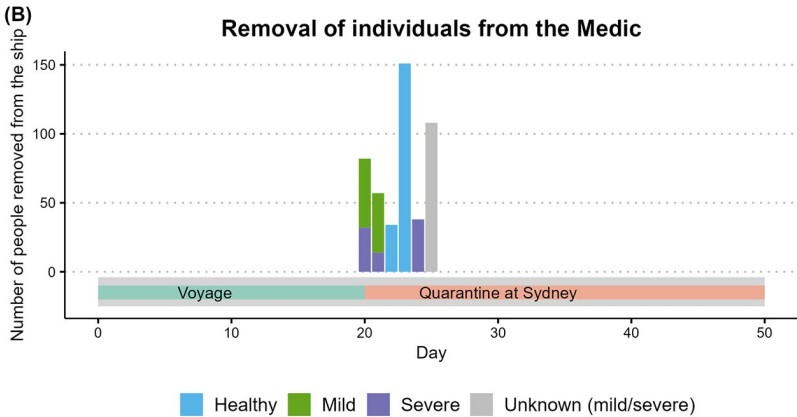

**(C)**

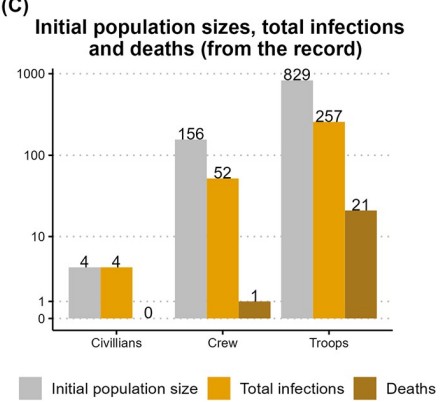

**Fig 2. A snapshot of data relating to the outbreak on board *Medic*.** Plot (A) displays the daily new severe infections among the ship's population during the voyage and quarantine period in Sydney. Plot (B) displays the removal of mild or severe infectious/ infected and healthy people on board *Medic* during the quarantine period. Plot (C) compares the total infections and deaths that occurred with respect to the initial population size by group.

## 2.3 Modeling disease outbreaks

To model the voyage of a ship, we considered the time the ship started its journey from the port of the presumed source of infection to the time when the number of infections on board reached zero (or almost zero). We modeled the dynamics of (1) new infections occurring from

the start of the journey to the arrival at an Australian Quarantine Station, and (2) the new infections occurring among the ships' personnel during the quarantine period. The latter type of dynamics may or may not have been observed depending on the ship. When the dynamics of the epidemic on a ship needed to be modeled during its time at the port of quarantine, we only considered the new infections that occurred on board the ship, disregarding the infections that took place in the quarantine station among the quarantine staff and among the healthy (who may have been exposed or pre-symptomatically infectious) people who had been removed from the ship for the purpose of isolation or quarantine.

During a ship's journey, we examined two groups: crew and passengers. Passengers may include civilians and/or troops. The groups may not have had identical mixing behaviour and their health conditions may have varied. We used stochastic metapopulation model structures to model the dynamics of the outbreaks where each group's model structure, but not parameters, is identical.

Let $N(t)$ be the total size of the population of the ship at time $t$ and $N_i(t)$ be the size of the $i$th group ($i$=1 (crew); $i$=2 (passengers)) at time $t$. We assumed that initially in Group $i$, individuals were fully susceptible ($S_i$) at the port of the presumed source of infection or where the vessel started its journey. Following exposure to an infectious individual, a susceptible person ($S_i$) may have become infected, entering the exposed ($E_i$) class, before becoming either asymptomatic ($A_{1i}$ and $A_{2i}$) or pre-symptomatic ($P_{ip}$), and then, symptomatic and infectious. A symptomatically infectious person may have suffered severe illness while infectious ($I_{iS}$) or mild illness while infectious ($M_i$), and following their infectious period, entered a symptomatic but non-infectious compartment ($C_i$ or $C_{Mi}$ respectively), before recovering ($R_i$). We assumed that all asymptomatic infections recovered ($R_{Ai}$). We also modeled deaths (denoted with parameter $d_i$) due to influenza occurring in $I_{iS}$ and $C_i$ compartments. The parameters relating to the transitions between the disease states and their explanations are included in S1 Text.

Apart from the transition between different disease compartments, we also modeled the removal of healthy people (in susceptible, exposed, asymptomatic, recovered, pre-symptomatic compartments at rates $\epsilon_i$) and cases (individuals in severely infectious, mildly infectious, or recovering compartments at rates $\zeta_i$). The number of individuals that were removed from the ship generally varied by day, and therefore, we assumed these parameters were time-dependent. In particular, these rates were zero while the ship was at sea and once the ship arrived at the quarantine station, we assumed that the removal parameters were constant throughout the day and changed with the start of the next day.

The record of *Medic* does not differentiate between those deaths that took place on board the ship and those deaths that took place after disembarkation from the ship. On the other hand, *Boonah*'s record implied that most of the deaths occurred at the Quarantine Station after the individuals had disembarked. We account for deaths both on and off the ship by modeling the disease dynamics of the severely infectious/infected ($I_{iS}$ and $C_i$ compartments) persons after they have been removed from the ship. An infectious person with severe illness ($I_{iS}$) who was taken to the Quarantine Station immediately entered the severely infectious and quarantined ($I_{iQ}$) compartment and then the quarantined symptomatic but non-infectious compartment ($C_{iQ}$) following their infectious period. Furthermore, a non-infectious person with severe illness ($C_i$) who was taken to the Quarantine Station immediately entered the $C_{iQ}$ compartment. A person in the $C_{iQ}$ compartment recovered ($R_{iQ}$). Death could take place on board the ship (from the $I_{iS}$ and $C_i$ compartments) and in the Quarantine Station (from the $I_{iQ}$ and $C_{iQ}$ compartments). S1 Text provides more details on how we simulated paths (using the Doob-Gillespie [25, 26] algorithm) with this model structure.

Our model structures for each group (passengers, crew) are shown in Fig 3. When modeling the dynamics of *Medic*, we used the full model structure (Fig 3: *Medic*) based on the

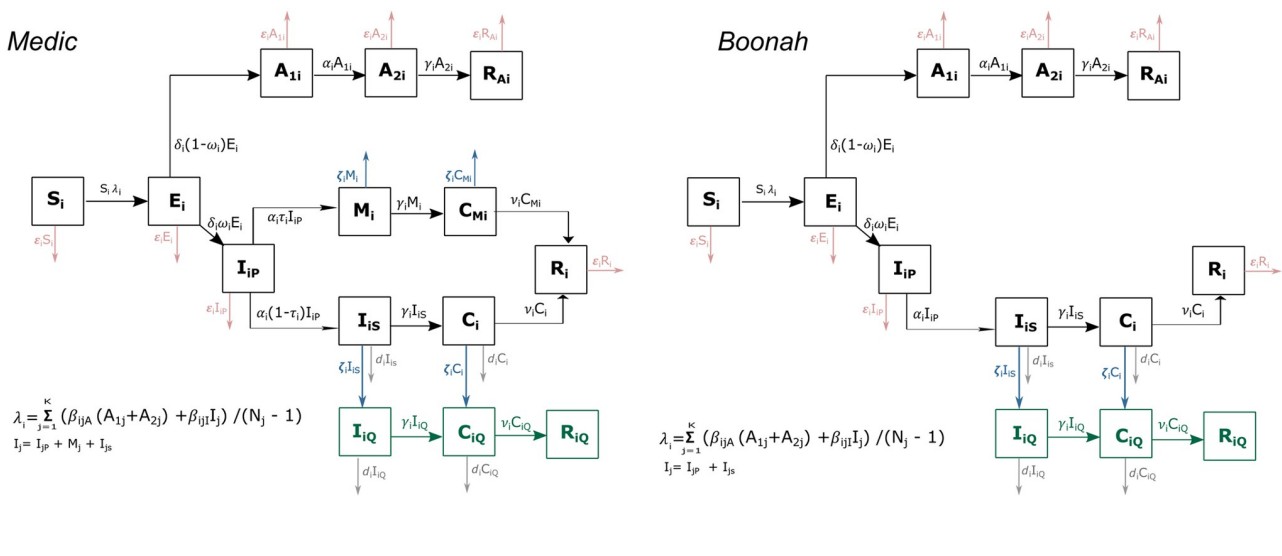

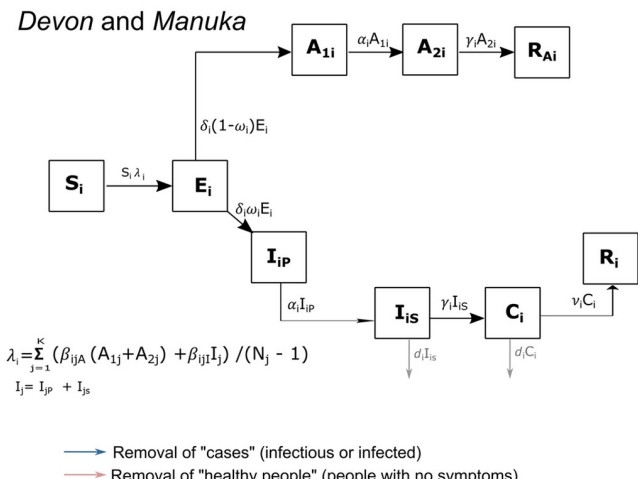

**Fig 3. Proposed models for *Medic, Boonah, Devon* and *Manuka*.** The compartments of the models are shown for group $i$ ($i=1$ (crew); $i=2$ (passengers)). The population is separated into those who are susceptible ($S_i$), exposed ($E_i$), asymptomatic ($A_{1i}$, $A_{2i}$), pre-symptomatically infectious ($I_{iP}$), mild ($M_i$) and severely ($I_{iS}$) infectious, mild and severely infected ($C_{Mi}$, $C_i$), severely infectious/infected and quarantined ($I_{iQ}$, $C_{iQ}$), and recovered ($R_{Ai}$, $R_i$, $A_{iQ}$). The force of infection, $\lambda_i$ has terms for both within- and between-group transmission.

detailed description of the outbreak in the original source material and our annotation of it (refer to S1 Text for details on the unsatisfactory performance of an alternative simpler model). This ensured that our model's complexity (in terms of the compartmental structure) was commensurate with the level of detail contained in the ship record, including for the removal of infectious, infected and healthy individuals. For *Boonah* (Fig 3: *Boonah*) and *Devon* and *Manuka* (Fig 3: *Devon* and *Manuka*), for which the original source material provides less information, we did not use the full model structure. Rather, we only included the flow through mildly infectious compartments as the records do not differentiate between severe and mild infections. Similarly, we did not model the removal of healthy, infectious, or infected individuals for *Devon* and *Manuka* as there were no records of removals for these ships. S1 Text describes the detailed dynamics for each of the ships and the corresponding modelling assumptions we made in relation to the available data.

## 2.4 Model assumptions

While addressing the challenges of the available data and constructing a viable parameter estimation process, we made the following model assumptions. We assumed that the rates of transmission from asymptomatic ($\beta_{ijA}$) and symptomatic infectious ($\beta_{ijI}$) are the same, denoted $\beta_{ij}$. We assumed that except for transmission parameters $\beta_{ij}$, all the other parameters across the groups were equal. We assumed that time-dependent parameters relating to the removal of healthy and infectious/ infected for a particular day were the same across both groups. We assumed that the initial conditions of the outbreaks were known. S1 Text details the assumptions made for the four ships and provides further explanations and details.

## 2.5 Estimation framework

We used a Bayesian hierarchical modeling framework [27–30] to study the outbreaks that took place on board the ships. Given that the ships sailed at similar times (October-November 1918) and that it is reasonable to assume that 1) all four ships were impacted by the same virus (the 1918 influenza virus); and 2) the demographic characteristics of those aboard the ships were similar, our choice of a hierarchical analysis is appropriate. Under this framework, we modeled each outbreak using the stochastic models introduced in Section 2.3. Each stochastic model was formulated as a continuous-time Markov chain with transition rates described in Fig 3. Under the hierarchical framework, we assumed that the transmission rates (between and within crew and passengers) are sampled from a common truncated multivariate normal distribution. We assumed that all the other model parameters were ship specific only. S2 Text provides details about the construction of the hierarchical model for the outbreaks. We used the two-step algorithm of [31] to estimate the parameters under the hierarchical framework. This algorithm makes use of Approximate Bayesian Computation (ABC) methods where the parameters are estimated by simulating paths from the model and matching them with observed data under specified distance criteria and tolerance values depending on the available data of the ships. S2 Text provides an explanation for the choice of prior distributions and describes the calibration of the algorithm. S2 Text presents the estimates of the parameters.

## 2.6 Studying the effects of quarantine measures

By the time *Medic* and *Boonah* arrived at Sydney and Fremantle Quarantine Stations respectively, a number of infectious and convalescent individuals were present on board. Furthermore, as a result of the quarantine measures at the time, infectious, convalescent, and healthy people were removed from the ship to the Quarantine Stations. We were interested in identifying if these interventions were beneficial in reducing the number of infections on board the vessel. We addressed this by posing two questions for each ship.

First, we asked by how much the interventions on the ships *Medic* and *Boonah* were estimated to have changed the final size of the epidemic. We addressed this by studying simulated paths that were close to the observed data which we will describe as conditional re-sampled paths. We generated these paths during the parameter estimation stage (Step 2 of the two-step algorithm which enables estimation of the ship-specific parameters under the hierarchical framework). For each proposed sample from the joint posterior, we repeatedly simulated paths along with the corresponding counterfactual path, along which there were no disembarkations, until a sample path (with disembarkations) satisfied the ABC acceptance/rejection criteria as used for the hierarchical estimation. We recorded the accepted samples from the joint posterior and the accepted paths along with their corresponding counterfactual paths to generate an estimate of the benefit (or otherwise) of the intervention.

Next, we asked if the disembarkations would have been expected to reduce the total infections on ships/voyages with properties as estimated for *Medic* and *Boonah*. The reason we were interested in this question is that in the records of [23], there were ships of similar characteristics as *Medic* and *Boonah*, but they could not be included in the analysis due to lack of sufficient data to model them. Therefore, to answer this question, we generated simulated paths with interventions and counterfactual paths from the joint posteriors. Unlike when addressing the first question, the ABC acceptance/rejection step was not applied. This method is similar to the posterior predictive check [32], and in this study, we will describe these simulated paths as re-sampled paths.

When the counterfactual with no disembarkation is considered, we simulated from the models for the *Medic* and *Boonah* with the rates of removal of healthy, infectious or infected set to zero. Further explanation on generating the conditional re-sampled and re-sampled paths is provided in S1 Text.

## 3 Results

Fig 4 illustrates and compares curve-wise intervals of the conditional re-sampled paths from the hierarchical estimation and curve-wise intervals of the re-sampled paths from the estimated parameters from the models in Fig 3 for the ships *Medic, Boonah, Devon,* and *Manuka*. See S2 Text for full trajectories of the conditional re-sampled and re-sampled paths for these ships.

The conditional re-sampled paths of *Medic* corresponded well with the observed data. As expected, the re-sampled paths, while in agreement with the observed data, have more variation than the conditional re-sampled paths.

While the general trajectories of the conditional re-sampled paths of *Boonah* were congruent with the observed data throughout the ship's journey, a notable difference is clear following arrival at the Quarantine Station, presumably due to some phenomena or activity that is not structurally captured in our model. For the conditional re-sampled paths and re-sampled paths, we calculated the median of the predicted peak day to be 13 [(12, 14) 25% and 75% quantiles] and 9 [(7, 11) 25% and 75% quantiles] respectively. The observed peak was on day 13. Within the observed outbreak, the infections on board started to appear after the fifth day of departure from the port of the presumed source of infection. However, even with our initial condition of one exposed person, the re-sampled paths implied that the outbreak most likely would have taken off as soon as the ship departed that port, and the outbreak most likely would have died out by the time the ship reached the Quarantine Station. Our hypotheses for these discrepancies in the conditional re-sampled and re-sampled paths are explored in the Discussion.

For *Devon*, the general trajectory for conditional re-sampled paths was in agreement with the observed data and the re-sampled paths have more variation. Furthermore, as time-series data for both crew and passengers for the ship were available, we were able to separately compare these paths with the observed data for the two groups (see S1 Text).

For *Manuka*, time-series data were only available after the ship arrived at the Quarantine Station, that is, from day 8. The day of the observed peak or the peak size is not available. From day 8, the conditional re-sampled paths corresponded well with the observed data and the re-sampled paths display more variation. Both types of re-sampled paths indicate that the epidemic likely took off as soon as *Manuka* left its port of the presumed source of infection. The median peak infection size was 8 [(5, 11) 25% and 75% quantiles] and 8 [(7, 9) 25% and 75% quantiles] for conditional re-sampled re-sampled paths respectively, and the median peak day

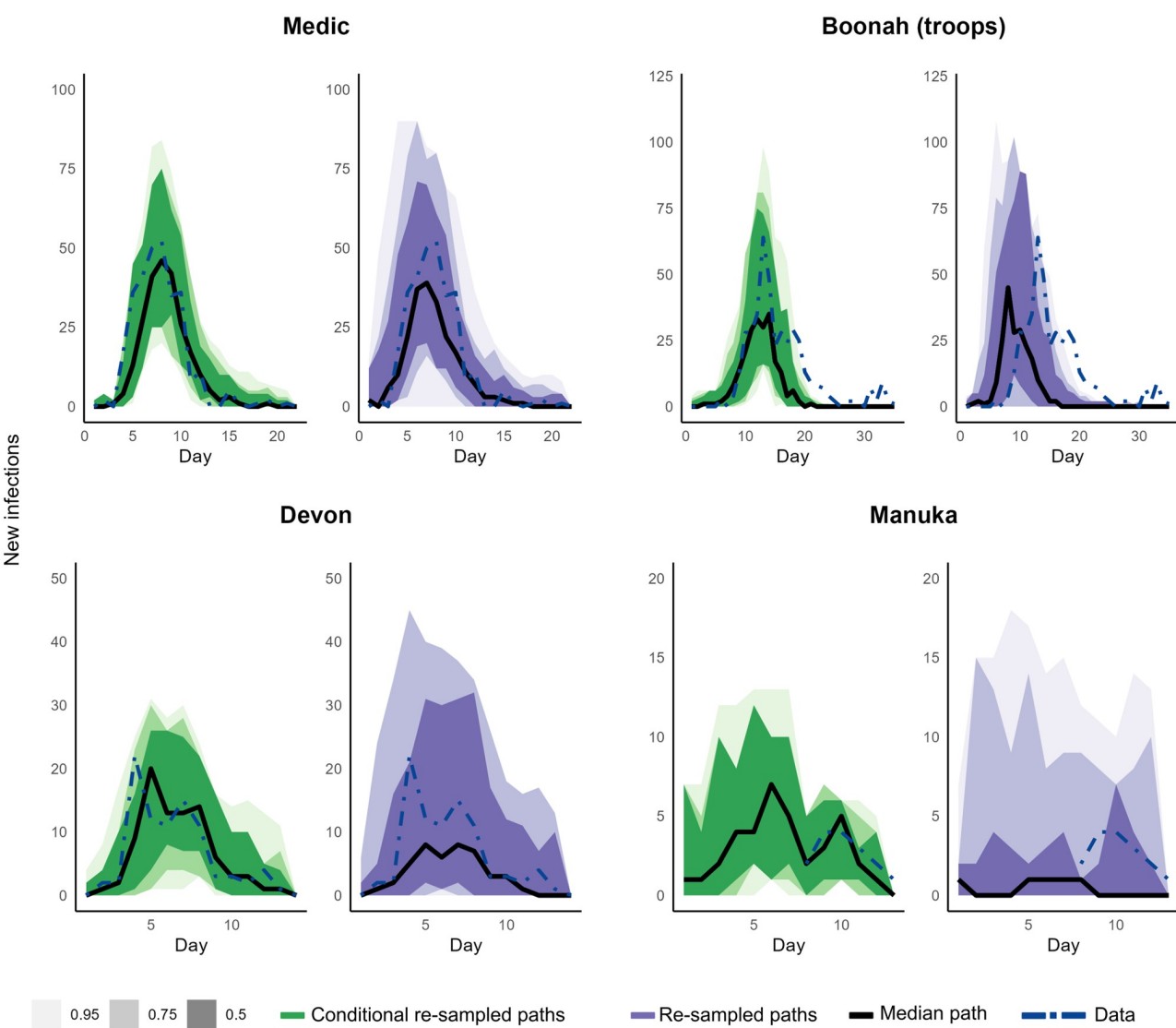

**Fig 4. Curve-wise intervals (50%, 75%, and 95%) of conditional re-sampled paths (in green) by the hierarchical estimation algorithm and re-sampled paths (in purple) by the estimated parameters.** Black lines are the median paths, and the blue dot-dashed lines represent the data.

was 5 [(4, 7.5)25% and 75% quantiles] and 5 [(4, 6) 25% and 75% quantiles] for conditional re-sampled and re-sampled paths respectively.

See S2 Text for time-series plots of infectious/infected and healthy individuals aboard *Medic* and *Boonah* based on re-sampled paths.

Fig 5 illustrates the posterior distributions of the transmission rates of the outbreaks. Table 2 shows the posterior medians and the corresponding 95% Highest Posterior Density (HPD) intervals. The transmission rates within the crew ($\beta_{CC}$) were similar among all four ships with a slightly higher transmission rate in *Manuka*. These posterior medians were 1.739 [(0.005, 4.551) 95% HPDI], 1.658 [(0.016, 4.200) 95% HPDI], 1.560 [(0.001, 3.488) 95% HPDI], and 1.969 [(0.431, 4.161) 95% HPDI] for *Medic, Boonah, Devon* and *Manuka* respectively. Furthermore, transmission rates from passengers to crew ($\beta_{CP}$) were very small (less than 0.6) and *Devon* had the smallest transmission rate [median 0.212 (0.003, 1.503) 95%

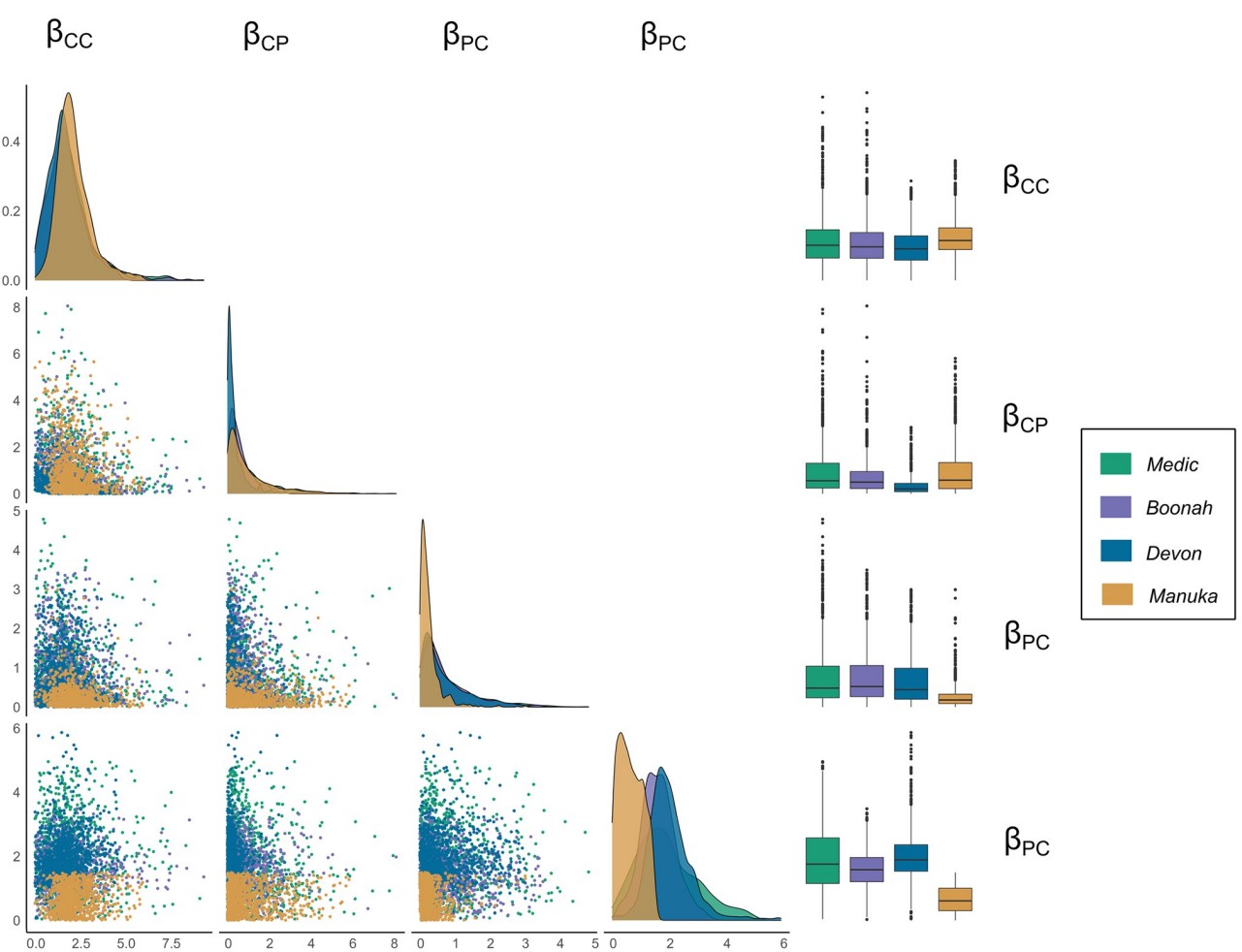

**Fig 5. Posterior distributions of the transmission rates under the hierarchical analysis. Diagonal:** The marginal posterior distributions. **Lower triangle:** Scatter plots between the transmission rates. **Right column:** Box-plots of the posterior distributions of the transmission rates.

HPDI]. Similarly, transmission rates from crew to passengers ($\beta_{PC}$) were small (less than 0.6) and *Manuka* had the smallest transmission rate [median 0.182 (0.004, 0.849) 95% HPDI]. The transmission rates within the passengers ($\beta_{PP}$) were higher for *Medic* [median 1.759 (0.072, 4.019) 95% HPDI], *Boonah* [median 1.582 (0.462, 2.887) 95% HPDI], and *Devon* [median

**Table 2. Posterior medians and 95% HPD intervals of the within and between transmission rates of crew and passengers in the ships.**

| Parameter | Medic | | Boonah | | Devon | | Manuka | |
|---|---|---|---|---|---|---|---|---|
| | Median | 95% HPD interval | Median | 95% HPD interval | Median | 95% HPD interval | Median | 95% HPD interval |
| $\beta_{CC}$ | 1.739 | (0.005,4.551) | 1.658 | (0.016,4.200) | 1.560 | (0.001,3.488) | 1.969 | (0.431,4.161) |
| $\beta_{CP}$ | 0.559 | (0.005,3.415) | 0.500 | (0.009,2.389 | 0.212 | (0.003,1.503) | 0.583 | (0.004,3.412) |
| $\beta_{PC}$ | 0.484 | (0.001,2.602) | 0.524 | (0.003,2.252) | 0.445 | (0.001,2.163) | 0.182 | (0.004,0.849) |
| $\beta_{PP}$ | 1.759 | (0.072,4.019) | 1.582 | (0.462,2.887) | 1.889 | (0.628,3.487) | 0.608 | (0.010,1.370) |
| | Median | Quartile | Median | Quartile | Median | Quartile | Median | Quartile |
| $R_0$ | 6.38 | (3.96, 11.06) | 2.53 | (2.07, 3.25) | 4.46 | (2.78, 8.97) | 2.45 | (1.87, 3.96) |

1.889 (0.628, 3.487) 95% HPDI], compared to that of *Manuka* [median 0.608 (0.010, 1.370) 95% HPDI]. Overall, transmission patterns aboard *Medic, Boonah,* and *Devon* were similar. Furthermore, transmission rates within a group were higher than the transmission rates from one group to another. S2 Text presents the estimates of the other model parameters.

We further estimated the basic reproduction number, $R_0$ from the estimated parameters (Fig 6). The calculation details for $R_0$ can be found in S1 Text. The *Medic* had the highest $R_0$ [median 6.38 (3.96, 11.06) quartiles]. The median of the estimated $R_0$ for *Devon* was 4.46 [(2.78, 8.97) quartiles], for *Boonah* was 2.53 [(2.07, 3.25) quartiles], and for *Manuka* was 2.45 [(1.87, 3.96) quartiles].

We then studied the impact of implementing quarantine measures on *Medic* and *Boonah* we described in Section 2.6.

### 3.1 *Medic*

Panels (A), (B), and (C) of Fig 7 illustrate the diagnostics performed to identify by how much the interventions on *Medic* changed the total number of infections. We addressed this question by studying the conditional re-sampled paths. If interventions had not taken place during the quarantine period of *Medic*, the median total number of infections throughout the epidemic would have increased by 18 [(5, 31) lower and upper quartiles] (see Table 3). The overall relative change in the increase of cases would have been 6.99% [(2.03, 11.49)% lower and upper quartiles]. This increment most likely would have been due to the infections that would have taken place among the passengers (that is, troops, see S1 Text for a breakdown of the cases). We calculated that a median of 2 [(-2,7) lower and upper quartiles] more deaths would have taken place if no interventions were implemented for the *Medic* (see S1 Text).

Panels (D), (E), and (F) of Fig 7 illustrate the diagnostics performed to identify if interventions would have reduced the number of infections for a ship that had similar properties to *Medic*, that is, another voyage. We tackled this question by studying the re-sampled paths. Similar to the previous setting, we did not observe a substantial increase in the total number of infections if no interventions were implemented for a ship that had similar properties as *Medic*. The median of the total infections would have increased by 11 [(1, 23) lower and upper quartiles], a relative increase of 4.54% [(1.14, 8.96)% lower and upper quartiles]. Furthermore, we evaluated that a ship with similar characteristics such as *Medic* would not have avoided any additional deaths [(-3,3) lower and upper quartiles] when interventions were in place (see S1 Text).

Note that the histograms (panels (C) and (F) of Fig 7 and the ones illustrated consequently), show some values for the difference in the total number of infections between paths under the main and counterfactual analyses to be less than zero, indicating that the counterfactual paths may have died out earlier or otherwise resulted in fewer infections. This is entirely expected due to the stochastic nature of the models.

### 3.2 *Boonah*

For *Boonah*, our analysis indicate that the measures had minimal influence. No substantial increase in infections is likely to have occurred had the quarantine station not implemented any interventions (see Fig 8 and Table 4). The relative change in cumulative cases ranged from 0% to 1.83%. Noting the expected early take-off in re-sampled paths, and so, minimal epidemiological activity by the time of arrival at the Quarantine Station, this percentage is estimated to be zero if a ship with properties similar to *Boonah* had travelled.

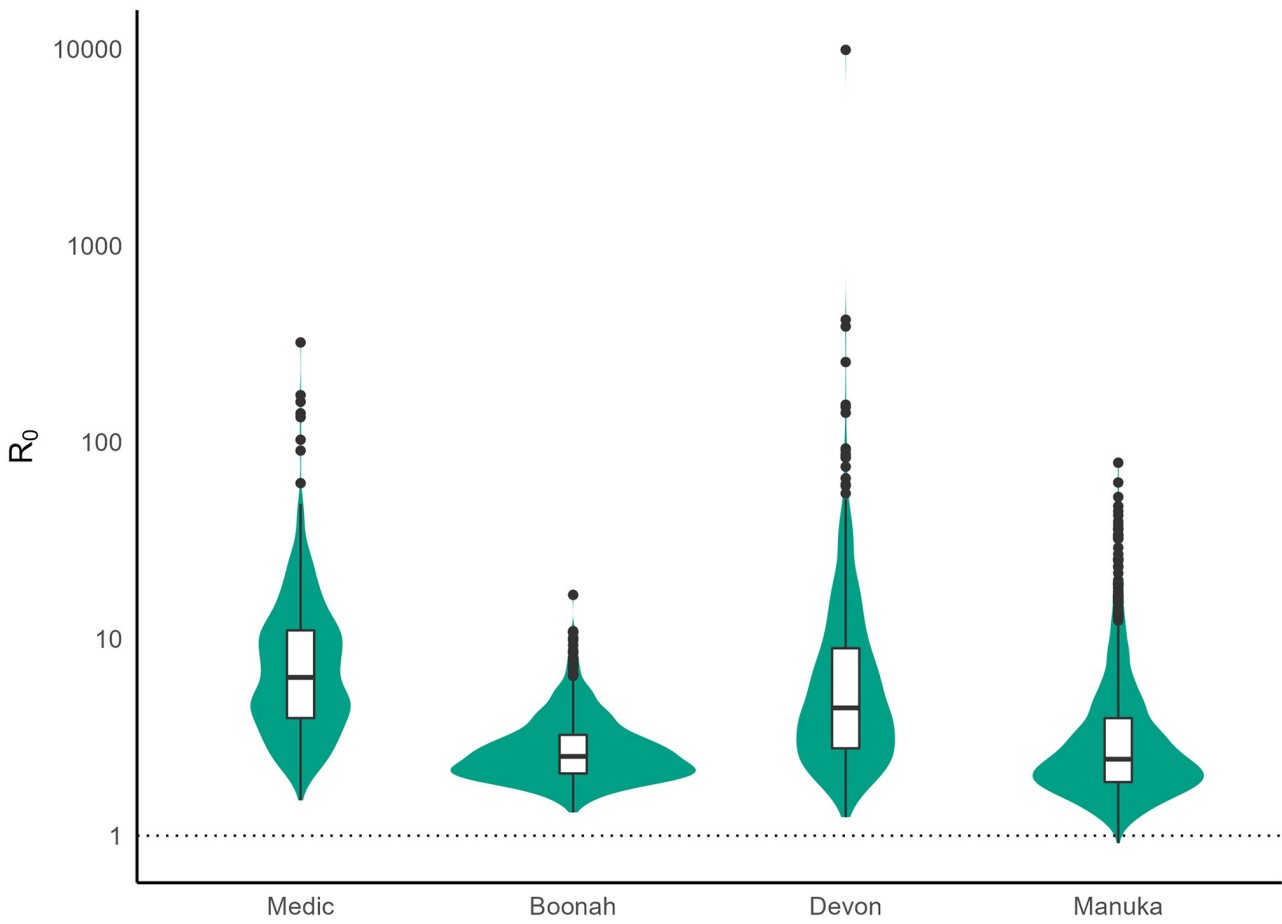

**Fig 6. Estimated $R_0$ for the ships.**

## 4 Discussion

We have studied the 1918 influenza outbreaks that took place on board the ships *Medic, Boonah, Devon* and *Manuka*.

We modeled each outbreak using a ship-specific two-group stochastic metapopulation model within a hierarchical statistical framework. The conditional re-sampled paths and re-sampled paths of *Medic, Devon,* and *Manuka* were comparable to the observed time-series data of the outbreaks. However, the conditional re-sampled paths of *Boonah* display a clear deviation from the data from following the 17th day, the day of arrival at the Quarantine Station. While this indicates that our model has almost certainly not captured all of the influences of disease transmission dynamics aboard the *Boonah* and/or the observation process for the outbreak, there was no further information within the historical record to justify introduction of a more complex model. We further note that once a ship arrived at the Quarantine Station, the ship's population was tested, and therefore, a greater fraction of infectious cases than what had been identified during the voyage may have been identified, increasing case ascertainment. These additional cases may also have included infectious cases with mild symptoms. This provides one plausible explanation for the observed deviation between the data and our model's output at the tail-end of the course of the epidemic. However, this hypothesised explanation

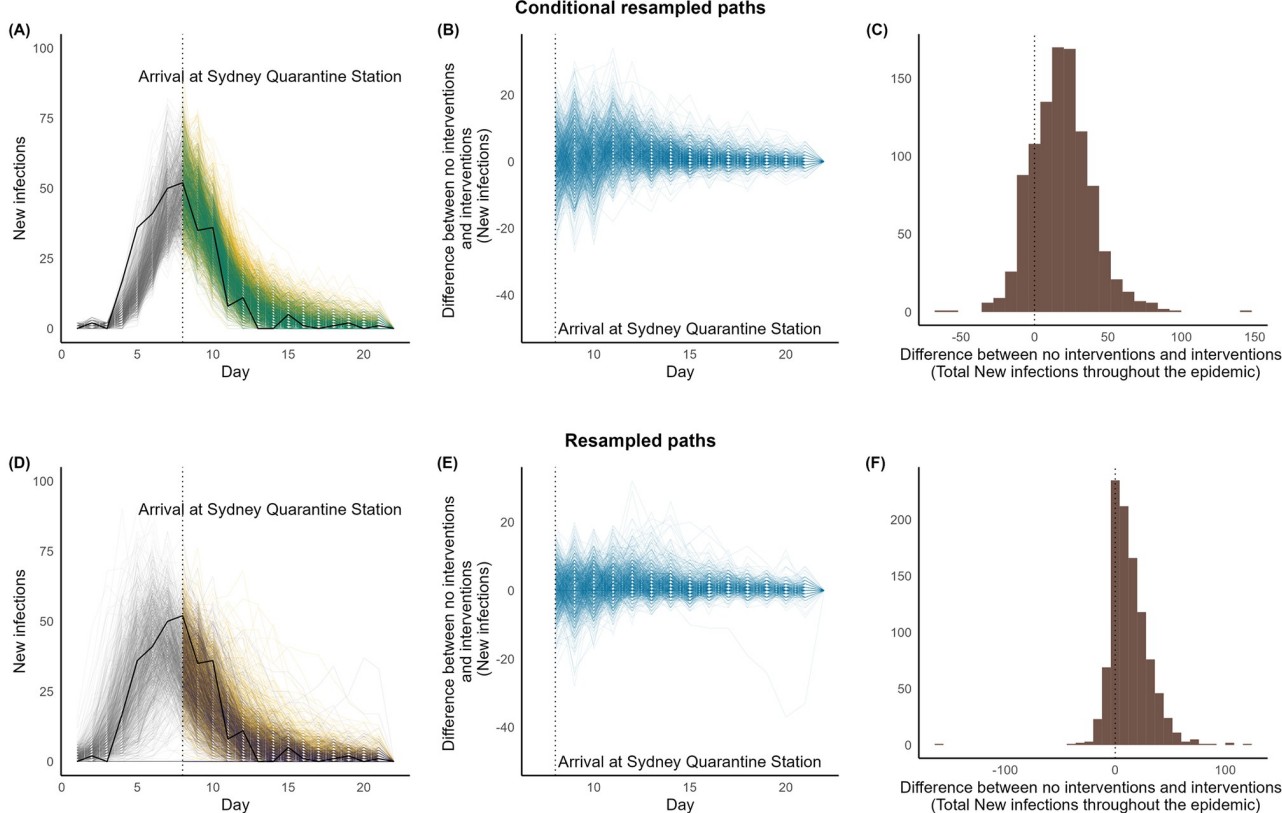

**Fig 7. Interventions vs. no interventions for *Medic* during their quarantine periods in Australia. Panels (A) and (D): Black solid line:** Observed time-series data. **Simulated paths in Grey:** Trajectories up to the time the ship arrived at the quarantine Station. **Simulated paths in Green:** Conditional re-sampled paths. **Simulated paths in purple:** re-sampled paths. **Simulated paths in yellow:** Counterfactual paths of no interventions corresponding to conditional re-sampled or re-sampled paths. **Panels (B) and (E):** Paths of the difference between no interventions and interventions. **Panels (C) and (F):** Difference in total new infections between no interventions and interventions.

cannot be evaluated as our model (appropriately) does not capture nor differentiate between mild and severely infectious persons since this aspect of the epidemic was not described in the record.

We estimated that the transmission within crew and passengers were higher than the transmission between crew and passengers. In ships such as *Medic, Boonah,* and *Devon* where the majority of the passengers were troops, the transmission within the crew and passengers were similar. However, *Manuka*'s transmission rate within the passengers was lower. Compared to the high population sizes in *Medic, Boonah,* and *Devon*, *Manuka*'s population size was smaller, specifically the number of passengers who were likely to be civilians. Therefore, less crowded settings aboard *Manuka* may have resulted in lower transmission among the passengers.

We estimated that the $R_0$ for the *Medic* and *Devon* were higher than *Boonah* and *Manuka*. The values of 6.38 (3.96, 11.06) for *Medic* and 4.46 (2.78, 8.97) for *Devon*, while large compared

**Table 3. Difference between no interventions and interventions (total infections throughout the epidemic) for *Medic*.**

|  | Median (25, 75)% | % of relative change Median (25, 75)% | Proportion of positive difference |
|---|---|---|---|
| **Conditional re-sampled paths** | 18 (5, 31) | 6.99 (2.03, 11.49) | 0.821 |
| **Re-sampled paths** | 11 (1, 23) | 4.54 (1.14, 8.96) | 0.759 |

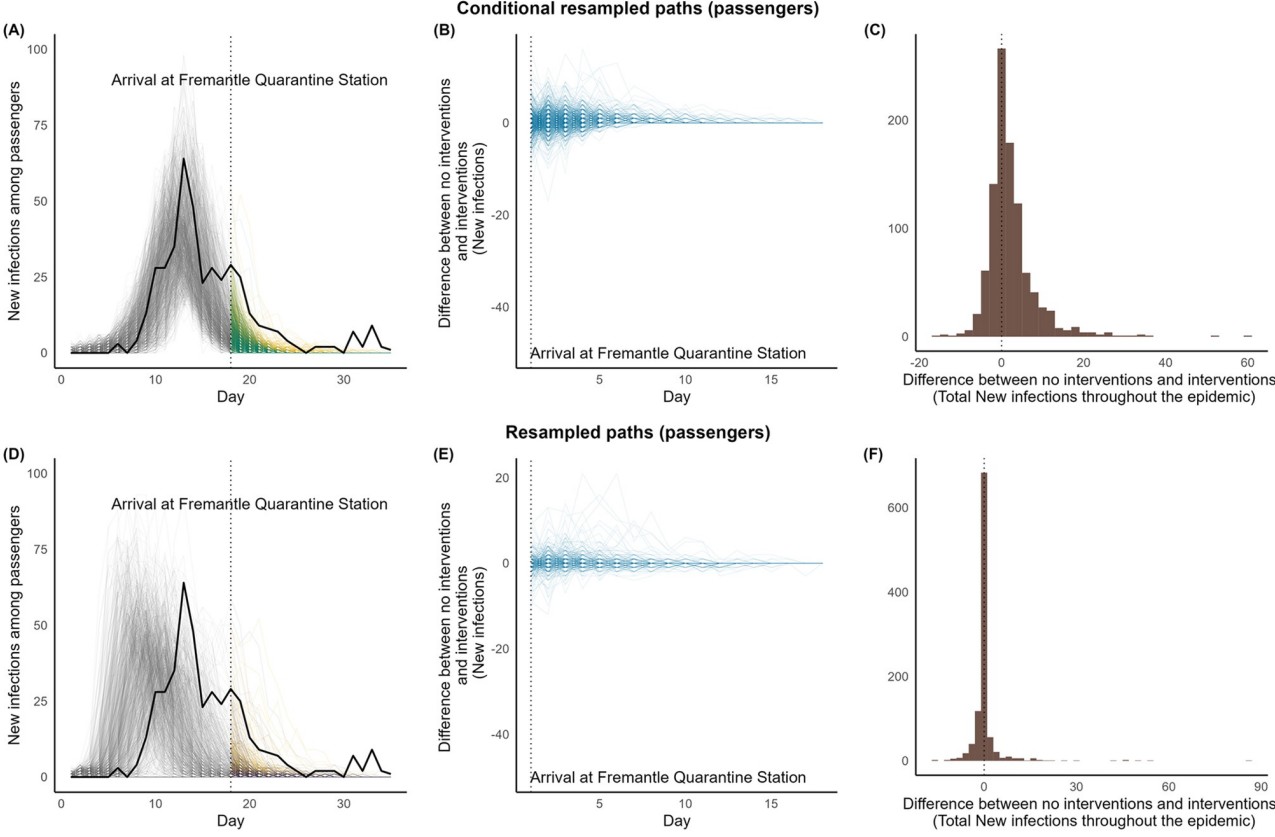

**Fig 8. Interventions vs. no interventions for *Boonah* during their quarantine periods in Australia. Panels (A) and (D): Black solid line:** Observed time-series data. **Simulated paths in Grey**: Trajectories up to the time the ship arrived at the quarantine Station. **Simulated paths in Green:** Conditional re-sampled paths. **Simulated paths in purple:** re-sampled paths. **Simulated paths in yellow:** Counterfactual paths of no interventions corresponding to conditional re-sampled or re-sampled paths. **Panels (B) and (E):** Paths of the difference between no interventions and interventions. **Panels (C) and (F):** Difference in total new infections between no interventions and interventions.

to population-level estimates of $R_0$ for the 1918 pandemic [33–36], are consistent with estimates from other closed settings, including boarding schools [13, 37, 38].

When modeling possible interventions for the outbreaks, we examined the interventions that took place once a ship reached a Quarantine Station, that is, the removal of infectious, convalescent, and healthy individuals. These interventions took place on board *Medic* and *Boonah*. For these two ships, we evaluated if these interventions were estimated to have been effective public health measures. For *Medic*, both for the actual voyage and possible voyages, we determined that the measures taken had a clear impact on the outbreak. If the interventions had not been put in place, approximately 7% and 4.5% more infections would have occurred respectively (Table 3). In contrast, for the *Boonah*, our analyses indicate that the impact was

**Table 4. Difference between no interventions and interventions (total infections throughout the epidemic) for *Boonah*.**

|  | Median (25, 75)% | % of relative change Median (25, 75)% | Proportion of positive difference |
| --- | --- | --- | --- |
| **Conditional re-sampled paths** | 1 (0, 5) | 0.53 (0, 1.83) | 0.629 |
| **Re-sampled paths** | 0 (0, 0) | 0 (0, 0) | 0.201 |

negligible. *Boonah* arrived at the Quarantine Station when the outbreak had passed its peak, and therefore the implementation of quarantine measures provided limited benefit.

The records concerning *Medic* and *Boonah* describe the presence of hospitals on board, and the adoption of early intervention measures such as adding isolation areas on board or using other decks to expand the hospital area, early detection of cases through temperature checking and restricting the interactions between the ships' population and the people on shore [23]. These factors may have contributed to controlling the epidemic to a greater extent and it may be of interest to quantify their effect on the progression of the epidemic by the time a ship arrived at a Quarantine Station. Unfortunately, data to capture these factors are limited.

The record of *Manuka* mentions that the removal of individuals did not take place at the Sydney Quarantine Station due to a lack of capacity. It was further recorded that all the unwell individuals were therefore initially isolated in their cabins. As the records did not further explain these dynamics, we were not able to model this to assess its impact.

We note that improved estimates and insights into the outbreaks on board ships during the 1918 influenza pandemic may be obtained if more outbreaks with suitable time-series data were studied. In the absence of that data, the only other potentially suitable outbreaks that may be of interest are those that took place onboard *Niagara, Nestor, Ceramic,* and *Makura* which provide only final size data or only group-aggregated time-series data. Further investigation of how inclusion of that additional data may help (or hinder) analysis is left as a topic for future study.

Another possible scenario that can be explored from this study is the impact on the Australian community if maritime quarantine measures had not taken place. In such a setting, one may need to be mindful of different mixing patterns among different communities/ states as well as the prevalence of infections in 1918–19 in Australia.

We employed a hierarchical estimation framework. This framework was instrumental in improving the overall parameter estimates in comparison to studying each outbreak independently (refer to S2 Text for a comparison of curve-wise intervals based on re-sampled paths and see [31] for an explanation). Through this study, we showed that when using a hierarchical framework, it is possible to capture the dynamics of each outbreak using a ship-specific stochastic model while still allowing for a common model structure across the outbreaks. Furthermore, while the record of *Manuka* had no time-series data until day 8, it does include the total number of infections that had occurred by day 8. The information sharing properties of the hierarchical modelling framework provided significant benefit here, leading to improved parameter estimates for *Manuka* (as seen by comparing independent and hierarchical estimation frameworks in S2 Text). We have assumed that the transmission rates have a hierarchical framework. Further improved parameter estimates may be obtained by constructing a full hierarchical model with all the model parameters that are common to the four outbreaks.

The data we studied were challenging in the sense that the records do not contain complete details of the dynamics of the outbreaks nor of interventions deployed. It may be of interest to apply this estimation framework to study the recent COVID-19 outbreaks that have taken place on board cruise ships.

Hierarchical models are, however, generally useful for improving parameter estimates, addressing missing and incomplete data issues, and allowing inferences on data at multiple levels (population and sub-population level) [28, 39, 40]. Furthermore, it is possible to use a hierarchical framework when the outbreaks are modelled as dynamical models [31, 41]. They have been previously applied to deterministic dynamical modeling settings in malaria [42]. However, computational time, efficiency and accuracy depend on the complexity of the dynamical model and the available data.

## Supporting information

**S1 Text. Data, modelling and assumptions.** Supporting information containing the details of data, codes, modeling and assumptions.
(PDF)

**S2 Text. Parameter estimation.** Supporting information containing the details of parameter estimation.
(PDF)

## Acknowledgments

Unless otherwise mentioned, computations were carried out in MATLAB or R across 32 clusters (32 virtual computers). All the computations were carried out by the use of the Nectar Research Cloud (project Infectious Diseases), a collaborative Australian research platform supported by the National Collaborative Research Infrastructure Strategy (NCRIS). All the plots were generated with ggplot2 [43] in R. The codes and necessary data are publicly available (see S1 Text for details).

## Author Contributions

**Conceptualization:** Punya Alahakoon, Peter G. Taylor, James M. McCaw.

**Data curation:** Punya Alahakoon.

**Formal analysis:** Punya Alahakoon, Peter G. Taylor, James M. McCaw.

**Investigation:** Punya Alahakoon.

**Methodology:** Punya Alahakoon.

**Software:** Punya Alahakoon.

**Supervision:** Peter G. Taylor, James M. McCaw.

**Validation:** Punya Alahakoon, Peter G. Taylor, James M. McCaw.

**Visualization:** Punya Alahakoon.

**Writing – original draft:** Punya Alahakoon.

**Writing – review & editing:** Punya Alahakoon, Peter G. Taylor, James M. McCaw.

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
