## [Decision Letter · Decision Letter 0]

8 May 2023

Dear Professor McCaw,

Thank you very much for submitting your manuscript "How effective were Australian Quarantine Stations in mitigating transmission aboard ships during the influenza pandemic of 1918-19?" for consideration at PLOS Computational Biology.

As with all papers reviewed by the journal, your manuscript was reviewed by members of the editorial board and by several independent reviewers. In light of the reviews (below this email), we would like to invite the resubmission of a significantly-revised version that takes into account the reviewers' comments.

In particular, please pay attention to Reviewer 1's first comment about the unclear motivation for the study, as well as two of the reviewers' comments about the over-complexity of the model and potential issues with identifiability.

We cannot make any decision about publication until we have seen the revised manuscript and your response to the reviewers' comments. Your revised manuscript is also likely to be sent to reviewers for further evaluation.

Sincerely,

Virginia E. Pitzer, Sc.D.

Section Editor

PLOS Computational Biology

Virginia Pitzer

Section Editor

PLOS Computational Biology

In particular, please pay attention to Reviewer 1's first comment about the unclear motivation for the study, as well as two of the reviewers' comments about the over-complexity of the model and potential issues with identifiability.

Reviewer's Responses to Questions

**Comments to the Authors:**

Reviewer #1: Please see attachment.

Reviewer #2: Thank you for the opportunity to review this manuscript. This manuscript looks at four outbreaks of Spanish flu on ships that were returning to Australia from WWI in 1918. The analysis estimates transmission rates amongst those on board and looks at whether maritime quarantine, implemented at that time, was effective (e.g. removing infectious, convalescent, healthy after arrival at quarantine station).

This manuscript is well written, the paper has a sensible and logical flow, the methods are reproducible, and conclusions follow from results. This work would be of interest to the epidemiological modelling readership and perhaps beyond given this is an interesting case study of historical data and deploys hierarchical ABC for parameter estimation. This was a pleasure to read and I have very few comments - I recommend this manuscript for publication as it has been submitted.

Some minor comments are below - I would suggest these minor changes to help the reader but I leave their inclusion to the authors’ discretion.

– Minor comments –

* Line 194: “rates of transmission between asymptomatic (βijA) and symptomatic infectious (βijI) are the same” -> perhaps rephrase to use “from” instead of ”between” since the transmission is between these and the susceptible class.

* Line 210: “sample paths” and “simulated paths” are used (L234) - consider aligning language throughout.

* Figure 7 : “qurantine” station

* Introduction: We demonstrated that generally, in the ships that transported troops, the transmission rates within crew and passengers were higher than those between the crew and passengers. -> suggest specifying “passengers (civilians and troops)”, otherwise it is lost on a reader how troops factor into the statement.

* Acknowledgements: please clarify “across 32 clusters”

* References: Several references need proper nouns capitalised (e.g. “Western Samoa”, “Spain”, etc).

* S2.1: 10. R_i = Recovered state of asymptomatic individuals. (I believe this is symptomatic individuals)

* S1: Can the perturbation kernel for the ship-specific ABCs be specified explicitly?

* S1: How were the tolerance values for the summary statistics determined?

* Code: I would suggest consolidating descriptions of the contents of each folder in the repo to all be in the main README file instead of distributed throughout the different folders.

* Code: I would suggest having a short description of how another researcher should run the code (i.e. very briefly state first, second, third steps; I believe all analyses are intended to be called from their respective folders).

* Code: It seems that all data necessary for reproducing the analysis is provided in the repository - it is worth stating this explicitly.

Reviewer #3: Uploaded as attachment.

**Have the authors made all data and (if applicable) computational code underlying the findings in their manuscript fully available?**

Reviewer #1: Yes

Reviewer #2: Yes

Reviewer #3: Yes

PLOS authors have the option to publish the peer review history of their article (what does this mean?). If published, this will include your full peer review and any attached files.

Reviewer #1: No

Reviewer #2: No

Reviewer #3: No
---

## [Decision Letter · Decision Letter 1]

2 Oct 2023

Dear Professor McCaw,

Thank you very much for submitting your manuscript "How effective were Australian Quarantine Stations in mitigating transmission aboard ships during the influenza pandemic of 1918-19?" for consideration at PLOS Computational Biology. As with all papers reviewed by the journal, your manuscript was reviewed by members of the editorial board and by several independent reviewers. The reviewers appreciated the attention to an important topic. Based on the reviews, we are likely to accept this manuscript for publication, providing that you modify the manuscript according to the review recommendations.

Please address the remaining minor comments raised by reviewer 3.

Sincerely,

Claudio José Struchiner, M.D., Sc.D.

Academic Editor

PLOS Computational Biology

Virginia Pitzer

Section Editor

PLOS Computational Biology

Please address the remaining minor comments raised by reviewer 3.

Reviewer's Responses to Questions

**Comments to the Authors:**

Reviewer #1: Thank you for your thoughtful responses. I have no further comments.

Reviewer #3: Thank you for (most of) the revisions/clarifications, they were indeed helpful since last reading this manuscript.

The remaining clarifications/edits are included for your consideration.

**Have the authors made all data and (if applicable) computational code underlying the findings in their manuscript fully available?**

Reviewer #1: Yes

Reviewer #3: Yes

PLOS authors have the option to publish the peer review history of their article (what does this mean?). If published, this will include your full peer review and any attached files.

Reviewer #1: No

Reviewer #3: No

Figure Files:

Data Requirements:

Reproducibility:

References:

---

## [Editor Report · Decision Letter 2]

3 Nov 2023

Dear Professor McCaw,

We are pleased to inform you that your manuscript 'How effective were Australian Quarantine Stations in mitigating transmission aboard ships during the influenza pandemic of 1918-19?' has been provisionally accepted for publication in PLOS Computational Biology.

Best regards,

Claudio José Struchiner, M.D., Sc.D.

Academic Editor

PLOS Computational Biology

Virginia Pitzer

Section Editor

PLOS Computational Biology

---

## [Editor Report · Acceptance letter]

16 Nov 2023

PCOMPBIOL-D-23-00364R2 

How effective were Australian Quarantine Stations in mitigating transmission aboard ships during the influenza pandemic of 1918-19?

Dear Dr McCaw,

I am pleased to inform you that your manuscript has been formally accepted for publication in PLOS Computational Biology. Your manuscript is now with our production department and you will be notified of the publication date in due course.

With kind regards,

Anita Estes
